# Long term term follow-up of tyrosine kinase inhibitors treatments in inoperable or relapsing diffuse type tenosynovial giant cell tumors (dTGCT)

**Mehdi Brahmi**[1], **Philippe Cassier**[1], **Armelle Dufresne**[1], **Sylvie Chabaud**[2], **Marie Karanian**[3], **Alexandra Meurgey**[3], **Amine Bouhamama**[4], **Francois Gouin**[5], **Gualter Vaz**[5], **Jerome Garret**[5], **Marie-Pierre Sunyach**[6], **Aurélien Dupré**[5], **Perrine Marec-Berard**[7], **Nadège Corradini**[7], **David Perol**[2], **Isabelle Ray-Coquard**[1,8], **Jean-Yves Blay**[1,8]*

1 Department of Medical Oncology, Léon Bérard Cancer Center, Lyon, France, 2 Department of Statistics, Léon Bérard Cancer Center, Lyon, France, 3 Department of Biopathology, Léon Bérard Cancer Center, Lyon, France, 4 Department of Radiology, Léon Bérard Cancer Center, Lyon, France, 5 Department of Surgery, Léon Bérard Cancer Center, Lyon, France, 6 Department of Radiotherapy, Léon Bérard Cancer Center, Lyon, France, 7 Institut d'Hematology Oncologie Pediatrique, Centre Leon Berard, Lyon, France, 8 University Claude Bernard, Lyon, France

* jean-yves.blay@lyon.unicancer.fr

**Data Availability Statement:** In the present study, we used patient data as source data and patients have not formally accepted to share their data on

## Abstract

### Rationale

CSF1R tyrosine kinase inhibitors (TKI) and antibodies yield response rates and tumor control in patients with diffuse type tenosynovial giant cell tumors (dTGCT). The long term management of patients with dTGCT treated with TKI is however not known.

### Patients and methods

We conducted a retrospective single center study on the 39 patients with advanced and/or inoperable dTGCT referred to the Centre Leon Berard for a medical treatment. The clinical characteristics and treatments of patients who had received at least one line of CSF1R TKI or Ab was collected from the electronic patient records and analyzed, after this study was approved by the Institutional Review Board of the Centre Leon Berard. Statistics were conducted using SPSS 23.0.

### Results

Thirty-nine patients received at least one line of TKI among the 101 patients with histologically confirmed dTGCT refered to this center. Imatinib, nilotinib, pexidartinib, emactuzumab were the most frequently used agents. First line treatment was given for a median duration of 7 months. With a median follow-up from the initiation of TKI of 30 months, the progression-free rate at 30 months is 56% for the 39 patients. 15 patients had recurrent disease after first line CSF1R inhibitor: 12 (80%) received a 2nd line treatment for a median duration of 6 months and a median time to progression (TTP) of 12 months. Six patients had

the web, even if fully anonymized. Therefore, the items of the dataset that we are able to share (no names, nodates, no identifier) are added as supplementary files.

**Funding:** JYB Institut National du Cancer, Direction Générale de l'Offre de Soins - NetSARC https://netsarc.sarcomabcb.org/ - RREPS https://rreps.sarcomabcb.org/ - RESOS sarcomes osseux http://www.infosarcomes.org/les-reseaux-netsarc-et-resos INCA-DGOS-INSERM 12563 : Site de recherche intégré en cancérologie LYRICAN INCa_4664 https://www.cancer-lyrican.fr/ INCA InterSARC http://www.infosarcomes.org/intersarc European Union Eurosarc (FP7-278742) http://eurosarc.eu/eurosarc/general-objectives/ EURACAN (EC 739521) European Reference Networks (ERNs) for adult rare solid cancers http://euracan.ern-net.eu/ Projets Investissements d'Avenir Laboratoire d'excellence (LabEx) DEvweCAN (ANR-10-LABX-0061) « Cancer, Développement, thérapies ciblées » http://devwecan.universite-lyon.fr/ Institut Convergence François Rabelais : Projet PLAsCAN (Prévenir la plasticité et l'adaptabilité tumorale : vers la nouvelle génération de médecine personnalisée) porté par l'Université de Lyon http://www.crcl.fr/75-Plasticite-des-cellules-cancereuses.crcl.aspx?language=fr-FR https://hub-recherche.fr/Pages/Recherche_news.aspx#k=plascan Association Association DAM's http://www.associationdams.org/association_dams_objectif_lutte_combat_mobilisation_sarcome.php Fondation ARC https://www.fondation-arc.org/ Infosarcome http://www.infosarcomes.org/ Ligue de L'Ain contre le Cancer http://www.liguecancer01.net/ La Ligue contre le Cancer https://www.ligue-cancer.net/ The funders had no role in study design, data collection and analysis, decision to publish, or preparation of the manuscript.

**Competing interests:** JYB: research support and honoraria from Novartis, Roche, Five Prime, Plexxikon, Daiichi Sankyo, Deciphera. MB, PC, AD, DP, SC Research support from Novartis, Roche, Five Prime, Plexxikon, Daiichi Sankyo, and Deciphera. This does not alter our adherence to PLOS ONE policies on sharing data and materials.

afterwards a recurrent disease and 5 (83%) received a 3rd line treatment for a median duration of 5 months and a median TTP of 9 months. Progression-free rate at 30 months was observed in 3 of 12 (25%) after line 2 and 1 of 5 (20%) after line 3. None of the patients refered died with a median follow-up of 67 months.

## Conclusions

CSF1R TKI or Ab provide prolonged tumor control and symptom relief for a majority of patients with inoperable or relapsing dTGCT, in first and subsequent lines. Multiple lines are required for close to 50% of patients with relapsing dTGCT.

## Introduction

Diffuse type tenosynovial giant cell tumors (TCGT) is a rare locally aggressive connective tissue tumor of the joints, affecting mostly young adults, with a predominance on the knee and ankle [1–4]. dTGCT frequently present at [1;2] translocation encoding for a fusion gene *CSF1/COL6A3* whose protein product plays a key role in tumor growth [5–7]. Surgical resection is the recommended treatment in first line [1–5].

However, local relapses frequently occur resulting in swelling, pain and functional impairement which are the hallmark of the disease [4,5]. Surgery is rarely curative after relapse with a 12% reported LRFS at 5 years in a large study [4]. Amputations may be required only very rarely in very large tumors; dTGCT metastasize very rarely [4].

Before CSF1R antagonists, either tyrosine kinase inhibitors (TKI) or antibodies, the medical treatments for relapsing and inoperable tumors had limited efficacy [1–3]. CSF1R antagonists have been reported to yield volumetric response and symptom relief in patients with inoperable diffuse type tenosynovial giant cell tumors (TCGT) [9–16]. Imatinib exerts CSF1R inhibitory activity, and was first reported as active in TGCT/PVNS in a case report in 2008 [9]. The clinical efficacy of tyrosine kinase inhibitors blocking CSF1R (imatinib, nilotinib, pexidartinib) and antibodies against CSF1R (emactuzumab, cabiralizumab) has been then confirmed in several retrospective clinical studies for imatinib [10,11], as well as prospective clinical trials, with emactuzumab [12], nilotinib [13], pexidatinib [14,15] and cabiralizumab [16].

Recently, Tap et al reported on a pivotal randomized phase III study comparing placebo with pexidartinib showing that tumor response was significantly higher with pexidartinib, and that patient reported outcome and function improved during treatment with pexidartinib as compared to placebo in this randomized double blind study [15]. Pexidartinib was recently approved for the treatment of dTGCT by the FDA. In addition to first demonstrate the clinical value of a TKI in this disease with unmet medical needs, this important study also proves that it is feasible to perform a randomized clinical trial in such a rare disease.

TKIs and Ab are administered during a limited period of time in all these studies, from few weeks to 12 months most often [8–16]. In the nilotinib phase II study, 30% of the patients stable after 12 months relapsed after nilotinib interruption, with 4 year PFS of 54% [13]. The impact of a retreatment with the same TKI or other CSF1R on dTGCT related pain and functional impairement has seldom been reported outside single cases [9,11].

Given the favorable life expectancy of these patients, it would be of importance to define a long term strategy for the medical treatment with CSF1R antagonists of patients with inoperable dTGCT treated with short term duration of TKI.

In the present work, we report a single center retrospective experience of the long term medical treatment of 39 advanced dTGCT, using sequential CSF1R antagonist treatments.

## Materials and methods

### Patients

Since Jan 2007, 39 patients referred to the Centre Leon Berard for a therapeutic decision for a dTGCT received a systemic treatment. These 39 patients represented 39% of the 101 patients with a central pathology confirmed dTGCT refered to the center during this time period. Central pathology review was obtained for all the patients, within the Reference pathology Centre of the Centre Leon Berard, according to the rules of the French NCI (INCa) with the NET-SARC [17,18]. The histological diagnosis of dTGCT was not confirmed in 16 of the 117 patients refered to the center during this time period. Giant cell tumor of the bone was the most frequent histological subtype for those unconfirmed dTGCT (not shown).

Table 1 describes the clinical characteristics of these 39 patients.

A retrospective collection of clinical history and treatment of these 39 patients was conducted, with the approval of the Institutional Review Board of the Centre Leon Berard (Comité de Revue des Etudes Cliniques, CREC, 28, rue Laennec 69008 Lyon on the date of Jan 19[th], 2019, Chair Dr Th. Bachelot), in addition to the data collected within the NETSARC and RREPS programs. Data on initial clinical presentation, past local and systemic treatments, response, outcome after treatment and present status of the patient were collected.

Treatment with TKI were given in 39 patients with tumors deemed inoperable and/or in whom surgery would not bring a clinical benefit. Treatment were given as part of a

**Table 1. Characteristics of patients treated with CSF1R inhibitors.**

|  | Mean (Range) | n (%) |
|---|---|---|
| **Gender** | | |
| Men | | 13 (33%) |
| Women | | 26 (67%) |
| **Age at diagnosis** (years) | 34.9 (13.2–59.3) | |
| **Age at TKI initiation** (years) | 40.4 (13.6–65.2) | |
| **Disease location** | | |
| Knee | | 17 (43.6%) |
| Ankle | | 9 (23.1%) |
| Foot | | 4 (10.3%) |
| Elbow | | 3 (7.7%) |
| Hip | | 2 (5.1%) |
| Wrist | | 2 (5.1%) |
| Hand | | 1(2.6%) |
| Finger | | 1 (2.6%) |
| **Previous surgeries for TGCT** | | 29 (74.4%) |
| **Time from diagnosis to CSF1Ri** (years) | 5 .5 (0.03–37.8) | |
| **Time from first surgery to CSF1Ri** (years) | 6.4 (0.7–37.8) | |
| **First line treatment** | | |
| Imatinib | | 15 (38.5%) |
| Nilotinib | | 4 (10.3%) |
| Emactuzumab | | 12 (30.8%) |
| Pexidartinib | | 2 (5.1%) |
| Other | | 6 (15.4%) |

compassionate off label use, or as part of clinical trials for experimental agents (NCT02371369, NCT01261429) which were previously published in peer reviewed journals [13,15,16]. The diagnosis of operability/non operability was taken by the weekly NETSARC multidisciplinary tumor board (MDT) dedicated to connective tissue tumors in place in the Centre Leon Berard, with a consensus obtained including 2 to 4 surgeons with expertise from connective tissue tumors. Generally, it was considered that surgery was the first treatment of choice if complete macroscopic resection of the tumor was deemed feasible for patients not previously operated. When complete macroscopic resection of the tumor was deemed not achievable at relapse (or would have required an amputation or a mutilating surgery not approved by the patient), patients were considered as "non-operable". The local extension of the disease, in these patients with mostly with multiple relapses, was often multifocal and involved both joints and tendon sheath.

### Statistics

Descriptive analysis of the patient population, and comparison between subgroups were performed using the IBM SPSS 23.0 package (IBM, Paris, France).

### Results

Table 1 presents the clinical characteristics of the 39 patients who have received a TKI, as compassionate use (imatinib, nilotinib,. . .), or within clinical trials for tyrosine kinase inhibitors or CSF1R Ab (Table 1). The median duration of the first line treatment for these 39 patients was 7 months (range 1–30+): 35 of these 39 (89.7%) of patients stopped the treatment for another reason than volumetric progression. With a median follow-up of 30 months since the initiation of first line TKI, 15 (38%) presented a novel volumetric progression and/or worsening symptoms, 11 after treatment discontinuation. Tumor progression was reported in 13 of 15 (87%) and worsening symptoms only in n = 2 (13%). Median time to progression (TTP) is not reached for these 39 patients: progression free rate was 56% at 30 months at the time of the analysis (Table 2 & Fig 1).

Twelve of these 15 (80%) patients restarted a TKI in second line, 11/12 (92%) because of volumetric progression with symptoms, 1/12 (8%) because of symptoms without volumetric progression. Median duration of treatment was 6 months (range 2–29). Median TTP was 12 months for a median follow-up of 29 months (Table 2, Fig 2). These 12 patients received the same (n = 2) or another (n = 10) CSF1R antagonist. 6 (50%) patients had progression and worsening symptoms after this second line, including 2 (16%) during second line treatment (Table 2).

Five of these 6 (83%) patients who received a second line treatment received a 3rd line treatment, for a median duration of 5 months (range 1–7) (Table 2) and with a median time to progression of 9 months (Table 2). 3 patients progressed (60%), 2 (40%) during treatment (imatinib & pazopanib for 1 patient each). Two patients received 4th line (both imatinib) and one received a 5th line treatment (sunitinib).

Fig 2 shows the duration of treatment and TTP for each individual patients and lines. Most patients had a prolonged TTP after treatment interruption. Duration of treatment was not correlated to TTP after interruption (not shown).

The compared duration of treatment, time to progression and time to next treatment according to the different agents used are presented in Table 2. The small number of patients and the retrospective nature of the study precludes the comparison of the outcome according to the different the TKI or Ab used. One patient each received twice imatinib and thrice imatinib, all progressing after the interruption of imatinib. The first one had a shorter duration of

**Table 2. Description of the lines of treatment.**

| | N | Duration of Treatment Months[1] | Volume Response N (%) | | | Symptom Improvement N (%) | Reason for interruption N(%) | | | TTP (median)[2] (TTP-range[5]) |
|---|---|---|---|---|---|---|---|---|---|---|
| | | | VR | SD | NE | | PD | AE [3] | Other[4] | |
| **Line 1** | | | | | | | | | | |
| *All* | 39 | 7.0 (1–30) | **13 (33)** | **20 (52)** | **6 (15)** | **24 (62)** | **2 (5)** | **12 (31)** | **25 (64)** | **56%* (3–29)** |
| *Imatinib* | 15 | 8.5 (1–30) | 3 (33) | 9(60) | 3 (20) | 11 (73) | 1 (7) | 3(21) | 11 (72) | 12 (3–14) |
| *Nilotinib* | 4 | 10.0 (7–12) | 0 | 4 (100) | 0 | 2 (50) | 1 (25) | 1(25) | 2 (50) | 17 (9–18) |
| *Emactuzumab* | 12 | 3.7 (1–8) | 4(33) | 6 (50) | 2 (17) | 7(68) | 0 | 4 (33) | 8(66) | 70%* (6–15) |
| *Pexidatinib* | 2 | 12(12–12) | 2(100) | 0 | 0 | 2 (100) | 0 | 0 | 2(100) | NA |
| *Others* | 6 | 6.7 (2–10) | 2(33) | 3(50) | 1 (17) | 3(50) | 0 | 3(50) | 3(50) | 12 (11–29) |
| **Line 2** | | | | | | | | | | |
| *All* | 12 | 6.1 (2–29) | **8 (66)** | **4(33)** | **0** | **10 (84)** | **2 (17)** | **3(25)** | **7(58)** | **12 (2–37)** |
| *Imatinib* | 7 | 7.8 (2–29) | 6 (86) | 1(14) | 0 | 6 (86) | 1 (14) | 3 (43) | 3(43) | 11 (6–13) |
| *Nilotinib* | 1 | 2 | 1 | 0 | 0 | 0 | 1 | 0 | 0 | 2 (n = 1) |
| *Emactuzumab* | 3 | 4(2–5) | 1(33) | 2(66) | 0 | 2 (66) | 0 | 0 | 3 | 37 (n = 1) |
| *Others* | 1 | 4 | 0 | 1(100) | 0 | 1 | 0 | 0 | 1 | no PD@ 8 months |
| **Line 3** | | | | | | | | | | |
| *All* | 5 | 5(1–7) | **2 (33)** | **3 (50)** | **0** | **3(50)** | **0** | **2 (40)** | **3(60)** | **9.0 (2–9)** |
| *Imatinib* | 3 | 5(2–7) | 1 (33) | 2 (66) | 0 | 2(66) | 0 | 1 (33) | 2 (66) | 9.0 (6–9) |
| *Pazopanib* | 1 | 1 | 0 | 1 | 0 | 1 | 0 | 1 | 0 | 2 (n = 1) |
| *Emactuzumab* | 1 | 6 | 1 | 0 | 0 | 0 | 0 | 0 | 1 | no PD @ 58 months |

1: Median (range)

2: Median (months) or *% progression-free at 30 months if median not reached

3: Adverse event

4: Other: patients in whom the treatment was not interrupted for progression or AE, or patients in whom treatment is still ongoing at the time of the analysis.

5: TTP-range in the subgroup of patients who reprogressed

NA: not applicable

VR: volumetric response.

efficacy in second line (9 vs 12 months) while the other had a similar duration of treatment efficacy.

## Discussion

Tyrosine kinase inhibitors of CSF1R have become the medical treatment of choice for relapsing or inoperable dTGCT, and a growing number of patients are receiving these treatments [9–16]. dTGCT is rarely a life-threatening disease. No deaths were observed in the 101 patients refered to us since 12 years. As a consequence, the majority of patients with inoperable dTGCT will be followed for a long period of time. The long term outcome of patients with dTGCT receiving TKI treatment has rarely been reported, even in prospective studies. The objective of the present study was to provide a description of the long term outcome of patients with dTGCT in the era of tyrosine kinase inhibitors which have been used for advanced dTGCT since 12 years in this center.

The present study shows first that the majority of the 39 patients with inoperable and/or relapsing dTGCT who received first line CSF1R antagonist have not progressed nor required additional treatment after first line treatment interruption during the observation period.

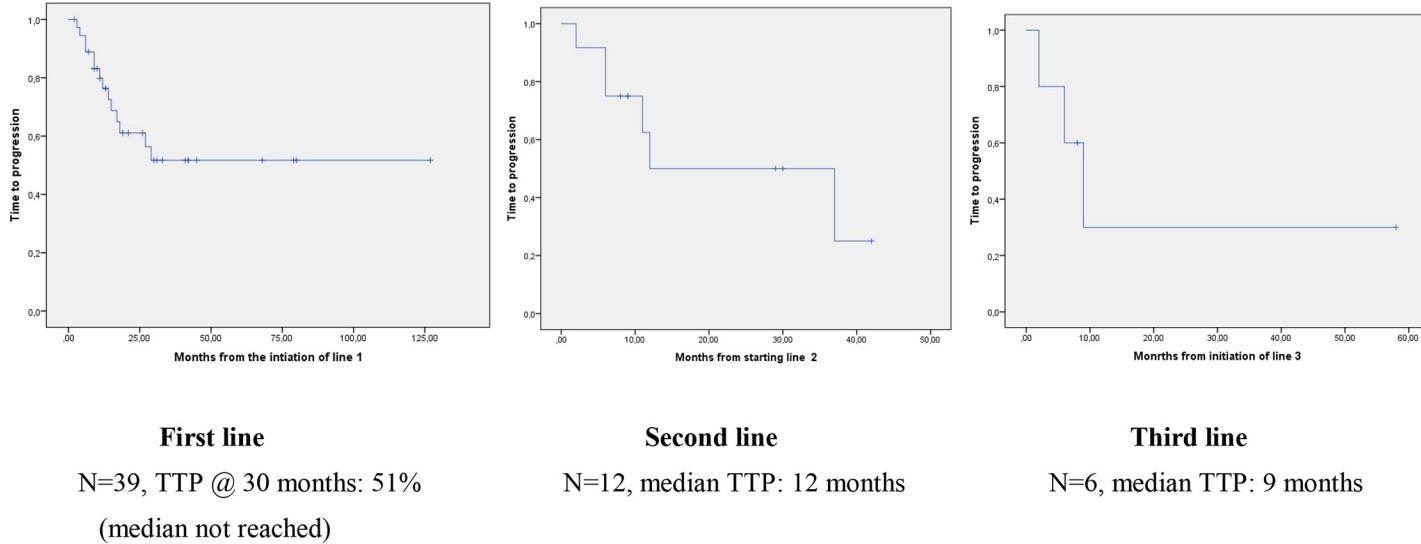

**First line**

N=39, TTP @ 30 months: 51%

(median not reached)

**Second line**

N=12, median TTP: 12 months

**Third line**

N=6, median TTP: 9 months

**Fig 1. Time to progression after line 1, 2, and 3.**

While first line median treatment duration was 7 months, the proportion of patients who have not progressed at 30 month is 56%, a proportion consistent with that reported in the previous prospective multicentric nilotinib phase II study [13]. The median follow-up of patients treated with first line TKI is however of 30 months only, and relapses at a longer term are likely to be observed, even though a plateau is observed after this date in this series.

A significant proportion of patients treated with a short term duration of TKI treatment do not progress rapidly after treatment interruption. Actually, four patients of this series are progression-free at 5 years after a duration of treatment inferior to 12 months. It is interesting to note that the majority of these patients still have clinically or radiologically detectable tumor.

Secondary volumetric and/or symptomatic progression occurred however in 15 patients, and 80% of these received a second line of CSF1R TK inhibitors. We used the terminology of line of treatment similar to that used for patients with advanced sarcoma receiving cytotoxics at each progression or symptoms requiring a novel treatment, even if the same treatment (eg imatinib) was reinstated for progression after discontinuation. The treatment was again given for a short period of time, shorter than in first line. However, again most patients benefited from this treatment, with a median time to progression of 12 months. While this is shorter than in first line treatment, 3 (25%) patients again reached a long TTP. These data show that second line CSF1R TKI are active in dTGCT, providing symptom relief and tumor control in a large proportion of patients.

Again, 6 (50%) of patients treated in second line progressed and became more symptomatic. These received a third line TKI, mostly with a different agent, within a clinical trial or as compassionate use. Treatment duration was again shorter in third line, with a median of 7 months, but median TTP was 9 months and 2 patients again benefited from long term tumor control.

The patients of this series received a variety of treatments either in clinical trials or in compassionate use. Similar or different treatment were used in the subsequent lines for the individual patients. A meaningful comparison of the impact of the different treatment is of course not possible in such series, but it will be of importance to report the long term outcome of patients included in clinical trials.

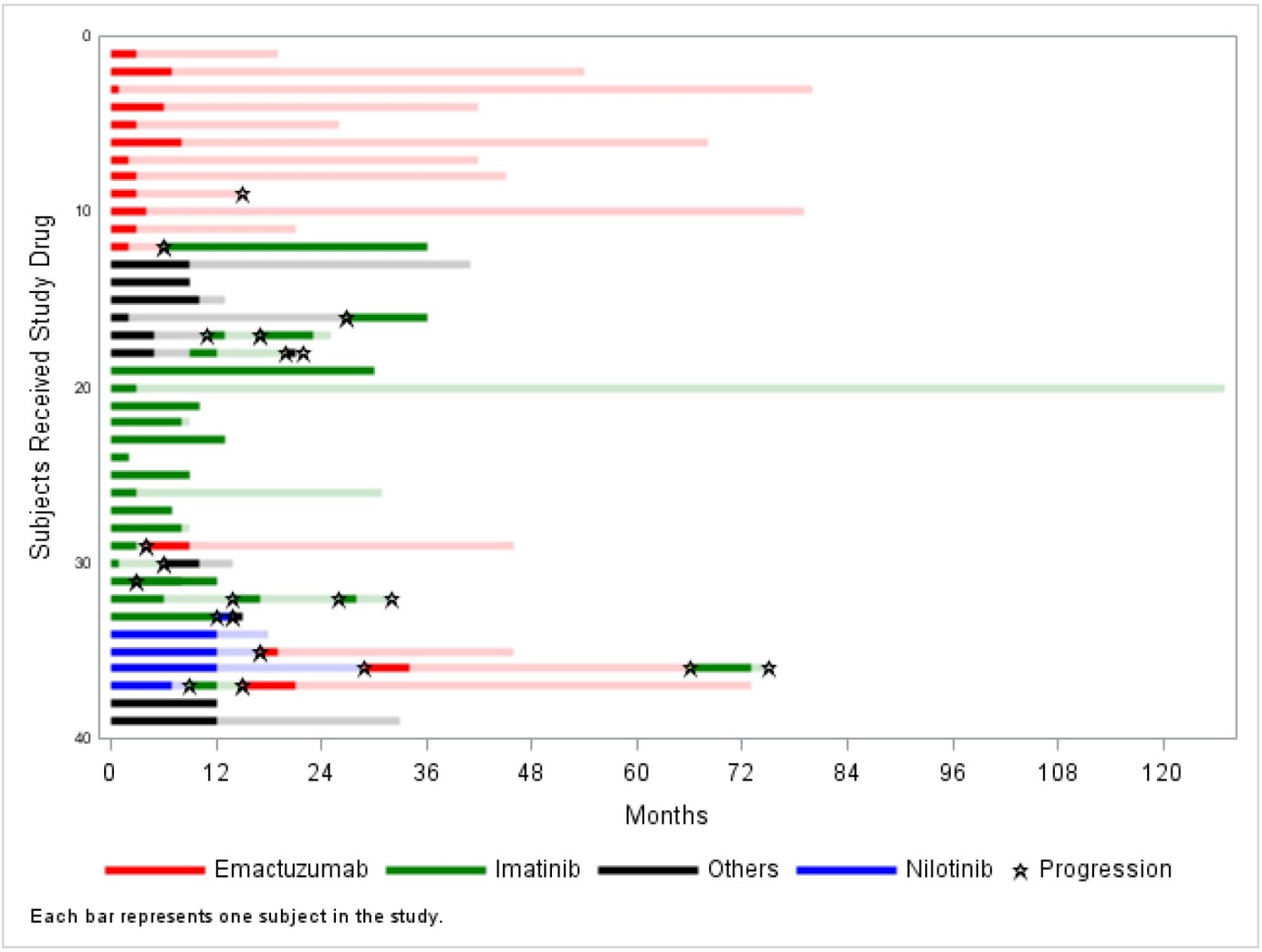

**Fig 2. Swimmer plot showing the duration of TKI treatment and TTP for the sequential lines in individual patients.**

There are several limitations in this study. First it is a single center, retrospective collection of patient observation over a 10 year period. A standardized description of symptoms and/or side effects leading to treatment decisions is lacking because of the retrospective nature of the study. The qualification of inoperable tumor in connective tissue tumors is debated, being related to both local extension, previous treatment in particular. In these patients, and as part of the standard procedure, a tumor is deemed inoperable after multidisciplinary evaluation in the multidisciplinary tumor board with at least 2 specialized surgeons. This is therefore a patient population identified by a multidisciplinary group of investigators specialized in connective tissue tumors. Importantly, only a minority of the patients referred for this team for medical treatment (39/101) were actually considered as inoperable and proposed for a medical treatment. Another limitation is that this retrospective study used an heterogenous set of CSF1R inhibitors with a broad (imatinib nilotinib.) or narrow (Abs, pexidartinib) spectrum of activity on tyrosine kinase; while comparison of agents is not feasible, it is still likely that future agents will be more often TKI or Abs with specific CSF1R inhibitory activities.

In this series, we observed that a short duration of administration of a CSF1R inhibitor can provide long term progression-free survival, after treatment interruption, in over 50% of patients with TGCT. Patients with recurrence of symptoms or progression may receive multiple lines of TKI. These treatments most often provide symptom relief and progression arrest, comparing favorably with those observed with secondary surgery [10]. The mechanisms and biomarkers predicting for long term control of locally TGCT treated with TKI are not known and deserve further investigation, considering in particular the molecular heterogeneity of this disease [19].

The value of secondary surgery has not been assessed in this study. A downstaging previous to surgical removal, instead of prolonged treatment could be a preferred option in relapsing dTGCT. Several considerations precluded the application of surgery in this period of observation: 1) most patients had minor shrinkage, 2) the uncertainty on the benefits of secondary surgical treatment, 3) the reluctance of many patients to undergo surgery in this situation of uncertainty. It is important to note that the benefit of secondary surgery was not obvious in an exploratory substudy of the nilogist trial [13].

In conclusion, relapsing or unresectable dTGCT is a rare neoplasia, for which a large number of active medical treatment providing tumor shrinkage and symptom relief are now described. First line TKI controls tumor progression in over 50% of the 39 patients at 30 months. For the remaining patients with relapsing disease after treatment interruption, the long term therapeutic strategies need to be defined taking into account the very favorable life expectancy of these patient and the functional consequences of tumor progression. Strategies involving short term treatment and drug holiday phases may deserve to be tested prospectively for this chronic disease.

## Supporting information

**S1 File. Unidentified data of patients included in this analysis.**
(DOCX)

## Author Contributions

**Conceptualization:** Gualter Vaz, David Perol, Isabelle Ray-Coquard, Jean-Yves Blay.

**Formal analysis:** Sylvie Chabaud, Jean-Yves Blay.

**Funding acquisition:** Jean-Yves Blay.

**Investigation:** Mehdi Brahmi, Philippe Cassier, Armelle Dufresne, Marie Karanian, Alexandra Meurgey, Amine Bouhamama, Francois Gouin, Gualter Vaz, Jerome Garret, Marie-Pierre Sunyach, Aurélien Dupré, Perrine Marec-Berard, Nadège Corradini, Isabelle Ray-Coquard, Jean-Yves Blay.

**Resources:** Aurélien Dupré.

**Supervision:** Jean-Yves Blay.

**Validation:** Jean-Yves Blay.

**Writing – original draft:** Jean-Yves Blay.

**Writing – review & editing:** Mehdi Brahmi, Philippe Cassier, Armelle Dufresne, Sylvie Chabaud, Marie Karanian, Alexandra Meurgey, Amine Bouhamama, Francois Gouin, Gualter Vaz, Jerome Garret, Marie-Pierre Sunyach, Aurélien Dupré, Perrine Marec-Berard, Nadège Corradini, David Perol, Isabelle Ray-Coquard, Jean-Yves Blay.

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
