## [Decision Letter · Decision Letter 0]

29 Oct 2019

PONE-D-19-26498

Long term term follow-up of tyrosine kinase inhibitors treatments in inoperable or relapsing diffuse-type tenosynovial giant cell tumors (dTGCT).

PLOS ONE

Dear Dr. Blay,

Thank you for submitting your manuscript to PLOS ONE. After careful consideration, we feel that it has merit but does not fully meet PLOS ONE’s publication criteria as it currently stands. Therefore, we invite you to submit a revised version of the manuscript that addresses the points raised during the review process.

We would appreciate receiving your revised manuscript by Dec 13 2019 11:59PM. To enhance the reproducibility of your results, we recommend that if applicable you deposit your laboratory protocols in protocols.io, where a protocol can be assigned its own identifier (DOI) such that it can be cited independently in the future. For instructions see: http://journals.plos.org/plosone/s/submission-guidelines#loc-laboratory-protocols

We look forward to receiving your revised manuscript.

Kind regards,

David M Loeb

Academic Editor

PLOS ONE

Journal Requirements:

2. In ethics statement in the manuscript and in the online submission form, please provide additional information about the patient records/samples used in your retrospective study. Specifically, please ensure that you have discussed whether all data/samples were fully anonymized before you accessed them and/or whether the IRB or ethics committee waived the requirement for informed consent. If patients provided informed written consent to have data/samples from their medical records used in research, please include this information.

3. Please note that all PLOS journals ask authors to adhere to our policies for sharing of data and materials: https://journals.plos.org/plosone/s/data-availability. According to PLOS ONE’s Data Availability policy, we require that the minimal dataset underlying results reported in the submission must be made immediately and freely available at the time of publication. As such, please remove any instances of 'unpublished data' or 'data not shown' in your manuscript and replace these with either the relevant data (in the form of additional figures, tables or descriptive text, as appropriate), a citation to where the data can be found, or remove altogether any statements supported by data not presented in the manuscript.

5.  Thank you for stating the following in the Financial Disclosure section:

JYB: research support and honoraria from Novartis, Roche, Five Prime, Plexxikon, Daiichi Sankyo, Deciphera. MB, PC, AD, DP, SC Research support from Novartis, Roche, Five Prime, Plexxikon, Daiichi Sankyo, and Deciphera.

We note that you received funding from a commercial source: Novartis, Roche, Five Prime, Plexxikon, Daiichi Sankyo, and Deciphera.

Reviewers' comments:

Reviewer's Responses to Questions

**Comments to the Author**

1. Is the manuscript technically sound, and do the data support the conclusions?

Reviewer #1: Partly

Reviewer #2: Partly

2. Has the statistical analysis been performed appropriately and rigorously? 

Reviewer #1: I Don't Know

Reviewer #2: Yes

3. Have the authors made all data underlying the findings in their manuscript fully available?

Reviewer #1: No

Reviewer #2: Yes

4. Is the manuscript presented in an intelligible fashion and written in standard English?

Reviewer #1: Yes

Reviewer #2: Yes

5. Review Comments to the Author

Reviewer #1: The authors present a retrospective review of patients treated with multiple tyrosine kinase inhibitors for dTGCT with long term follow up. They present the data and base their conclusion in terms of progression, the line of therapy used, and the medications employed. I am not aware of a similar review at this time and in this regard, feel it offers some valuable information for clinicians and patients.

I have a few questions or concerns.

1. Although the authors state that inclusion criteria included non-operable or relapse disease - they did not describe their rationale for inclusion with any other details. Non-operable disease can be very surgeon-dependent and may cover a broad spectrum. Similarly, recurrence is fairly common and one recurrence following inadequate surgery is very different from 4 recurrences after extensive surgical intervention.

2. Since the most common location is the knee joint and since the ultimate sequel is degenerative joint disease, I would expect that some of these patients would have gone on to joint replacement surgery or at least would be evaluated in terms of joint pain. There is very little information regarding the type of limitations or discomfort and other methods of management.

3. Joint disease is often surgically different from disease arising from the tendon sheath - I am not sure this distinction was made either. It has some implications in terms of surgical approach, ability to access or visualize disease, and ability to debride.

4. While non-specific drugs ie: imantinib initiated the interest in targeted therapy - it is likely more relevant to know the impact of CSF1R specific therapy. I imagine most clinicians would prefer to start with a specific and narrowly targeted agent, rather than a broader or less specific - even if historically these were used. I think the discussion may need to address the fact that this is an evolving landscape and that some of the treatment modalities will be less relevant but still perhaps lend interesting insight.

5. I wonder why there is no mention of using the medication to downstage patients from "in-operable" to "operable"? In other words - were any of the patients more amendable to surgical resection after medical management, were they not, and why? Was this a consideration at the time? I think the relevance is that this is preferably a surgical disease - and most imagine that indefinite medical management will be less preferred.

6. While including many different locations, patients, and medical treatments increases the cohort size - I wonder if it provides instructive information for clinical application. For example - it may be more useful for us to know that in patients with 2 recurrences - following reasonable surgical efforts, use of a CSF1R specific drug resulted in durable PFD for x number of months. This is hard to do with so many different medications, patients, tumors, locations, etc. At a minimum should be discussed.

7. No real limitations to the study are presented or discussed.

8. Some grammatical and stylistic issues need to be addressed. For example - on line 73 it reads "...the treatment of dTGCT by the." The sentence is unfinished.

Reviewer #2: This is a highly relevant topic, with a nice review of a single center's experience with anti-CSF1R treatment for diffuse tenosynovial giant cell tumor. This is an interesting and clinically applicable study population, though the variability of treatments, durations of treatments, and differences in context (compassionate use versus clinical trial) can confuse the interpretation of the findings. Some specific comments are included below:

Abstract -- With a median follow-up of 30 months, the time to progression (TTP) is 56% at 30 months. That is not a time to progression. Please clarify. I think I understand the authors' meaning, but it reads awkwardly.

"Sequential therapeutic strategy should be explored in patients with multiple relapses." I am not sure that the results and findings of this retrospective analysis of only patients on treatment, including only 39 patients, with various therapies, mostly with short followup, is particularly strong in supporting this statement.

The duration of treatment for many of these patients appear to be dictated by clinical trial. Perhaps this is an inaccurate assumption. However, that limits conclusions that can be drawn regarding duration of treatment.

Some spelling errors were identified (line 136, for example). Not sure that “reprogress” is a word in the Oxford English Dictionary (line 209)

In reality, this is a study of 39 patients treated with anti-CSF1R therapy, rather than the 101 suggested, with a median duration of 7 months. There is no clear indication for consideration of the other 62 patients who were not on treatment. There is no comparison of the anti-CSF1R group to those off treatment, so the abstract and manuscript should really more accurately reflect that this is a study of 39 patients.

If 15 patients progressed, but only 4 stopped first line therapy for progression, it begs the question: why didn’t the other patients with progression stop first line treatment for that reason?

Table 2 – combining “Adverse Events” with "patient request" makes conclusions difficult; please also clarify “scheduled treatment discontinuation or ongoing treatment” – as this would seem as though patients with ongoing treatment will be included in the reasons for discontinuation. The numbers don’t seem to reflect that, though. Please clarify.

Line 183 – Please clarify how the two patients who started the same CSF1R inhibitor as second line therapy for either progression with symptoms or symptoms without progression would still count as second line therapy. At first glance, that would seem to be continuation on therapy if the antagonist is the same. Presumably, these are patients who completed treatment, demonstrated progression off treatment, and were re-started on therapy again. To include this scenario (relapse after completion of therapy) together with patients who progressed on therapy really confuses the overall analysis. This is a large limitation of the current study.

6. PLOS authors have the option to publish the peer review history of their article (what does this mean?). If published, this will include your full peer review and any attached files.

Reviewer #1: No

Reviewer #2: No

---

## [Author Response · Author response to Decision Letter 0]

9 Jan 2020

Reviewer #1: The authors present a retrospective review of patients treated with multiple tyrosine kinase inhibitors for dTGCT with long term follow up. They present the data and base their conclusion in terms of progression, the line of therapy used, and the medications employed. I am not aware of a similar review at this time and in this regard, feel it offers some valuable information for clinicians and patients.

I have a few questions or concerns.

1. Although the authors state that inclusion criteria included non-operable or relapse disease - they did not describe their rationale for inclusion with any other details. Non-operable disease can be very surgeon-dependent and may cover a broad spectrum. Similarly, recurrence is fairly common and one recurrence following inadequate surgery is very different from 4 recurrences after extensive surgical intervention.

Answer: We fully agree with this comment. The definition of operability varies considerably according to the surgeon for connective tissue tumors, including across reference centers. In this single-center retrospective study, the diagnosis of operability/non operability was taken by the MDT in place on the site, with a consensus always obtained from 2 to 4 surgeons with expertise from connective tissue tumors. It was considered that surgery was the first treatment of choice in complete macroscopic resection of the tumor was deemed feasible for patients not previously operated. When complete macroscopic resection of the tumor was deemed not achievable at relapse (or would have required an amputation), patients were considered as non-operable. This was added in the material and methods section, chapter description of the patients, in this revised version. Please see page 4.

2. Since the most common location is the knee joint and since the ultimate sequel is degenerative joint disease, I would expect that some of these patients would have gone on to joint replacement surgery or at least would be evaluated in terms of joint pain. There is very little information regarding the type of limitations or discomfort and other methods of management.

Answer: This is an important point indeed. Because of the retrospective nature of the study, it can not be claimed that an exhaustive collection of symptoms was performed: pain, swelling, functional impairment were most often reported. The decision of treatment with TKI was discussed and taken only if clinical symptoms significantly impaired patient life and joint function, and if requested by the patient. Knee replacement was considered for some patients but not performed in any of these patients treated with TKI (yet). This was added in the discussion section. Please see page 7 of this revised version.

3. Joint disease is often surgically different from disease arising from the tendon sheath - I am not sure this distinction was made either. It has some implications in terms of surgical approach, ability to access or visualize disease, and ability to debride.

Answer: We also agree with this comment. The local extension of the disease in these patients who mostly have multiple relapses, was often multifocal and involved both joints and tendon sheath. This was added in the Material and Methods section of the revised version.

4. While non-specific drugs ie: imatinib initiated the interest in targeted therapy - it is likely more relevant to know the impact of CSF1R specific therapy. I imagine most clinicians would prefer to start with a specific and narrowly targeted agent, rather than a broader or less specific - even if historically these were used. I think the discussion may need to address the fact that this is an evolving landscape and that some of the treatment modalities will be less relevant but still perhaps lend interesting insight.

Answer: We agree with this comment. The text was amended accordingly, and now includes a new paragraph in the Discussion section.

5. I wonder why there is no mention of using the medication to downstage patients from "in-operable" to "operable"? In other words - were any of the patients more amendable to surgical resection after medical management, were they not, and why? Was this a consideration at the time? I think the relevance is that this is preferably a surgical disease - and most imagine that indefinite medical management will be less preferred.

Answer: we fully agree with this: a downstaging previous to surgical removal, instead of prolonged treatment should be the preferred option in such a chronic disease/tumor. Several considerations precluded the application of such surgery in this period of observation: 1) most patients had minor shrinkage, 2) the uncertainty of the benefits from secondary surgical treatment, 3) the reluctance of many patients to undergo surgery in this situation of uncertainty. It should be mentioned that the benefit of secondary surgery was not obvious in an exploratory sub-study of the niloGIST trial (Gelderblom et al. Lancet Oncol 2018), as now quoted in the discussion. This topic was expanded in the Discussion section. This important question needs to be addressed in a future study. 

6. While including many different locations, patients, and medical treatments increases the cohort size - I wonder if it provides instructive information for clinical application. For example - it may be more useful for us to know that in patients with 2 recurrences - following reasonable surgical efforts, use of a CSF1R specific drug resulted in durable PFD for x number of months. This is hard to do with so many different medications, patients, tumors, locations, etc. At a minimum should be discussed.

Answer: we agreed with this limitation. The rarity and intrinsic heterogenous nature of the clinical presentations of dTGCT makes it difficult to achieve homogenous population. A prospective collection would be required. Considering the rarity of the disease, we nonetheless felt that the real-life setting of this relatively large series, could guide the preparation of future prospective studies. We consequently clarified the limits of the study in a specific chapter of the discussion section in this revised version. 

7. No real limitations to the study are presented or discussed.

Answer: This is in the same line of thought as the comment above. We added a specific section on these questions at the end of the revised discussion.

8. Some grammatical and stylistic issues need to be addressed. For example - on line 73 it reads "...the treatment of dTGCT by the." The sentence is unfinished.

Answer: Please accept our apologies for this, we made the appropriate corrections.

 

Reviewer #2: This is a highly relevant topic, with a nice review of a single center's experience with anti-CSF1R treatment for diffuse tenosynovial giant cell tumor. This is an interesting and clinically applicable study population, though the variability of treatments, durations of treatments, and differences in context (compassionate use versus clinical trial) can confuse the interpretation of the findings. Some specific comments are included below:

Abstract -- With a median follow-up of 30 months, the time to progression (TTP) is 56% at 30 months. That is not a time to progression. Please clarify. I think I understand the authors' meaning, but it reads awkwardly.

Answer: we agree with this comment. We changed the text accordingly in the revised version: «With a median follow-up from the initiation of TKI of 30 months, the progression-free rate at 30 months is 56% for the 39 patients. » 

"Sequential therapeutic strategy should be explored in patients with multiple relapses." I am not sure that the results and findings of this retrospective analysis of only patients on treatment, including only 39 patients, with various therapies, mostly with short followup, is particularly strong in supporting this statement.

Answer: We agree with this comment, linked to the limitations of this study, and have downstated this statement in the revised version: « Sequential therapeutic strategy may need to be explored in patients with multiple relapses.»

The duration of treatment for many of these patients appear to be dictated by clinical trial. Perhaps this is an inaccurate assumption. However, that limits conclusions that can be drawn regarding duration of treatment.

Answer: This is indeed the case, the duration of treatment varied according to the context in which the treatment was given, with generally stringent and precise criteria for patients included in clinical trials, and more flexible approaches for off-label use. For this reason, no strong statements can be made on the optimal duration of treatment. This has been added in the discussion section

Some spelling errors were identified (line 136, for example). Not sure that “reprogress” is a word in the Oxford English Dictionary (line 209)

Answer: we made the corrections in the revised manuscript.

In reality, this is a study of 39 patients treated with anti-CSF1R therapy, rather than the 101 suggested, with a median duration of 7 months. There is no clear indication for consideration of the other 62 patients who were not on treatment. There is no comparison of the anti-CSF1R group to those off treatment, so the abstract and manuscript should really more accurately reflect that this is a study of 39 patients.

Answer: We agree that this is a study focusing on the 39 patients. We considered that it was important to mention that only a minority (39/101) of patients referred to us for a dTGCT were finally proposed for a systemic treatment. We detailed this in the abstract and in the text, and also added information on follow-up for the remaining 62 patients in the first chapter of the result section of this revised version.

If 15 patients progressed, but only 4 stopped first line therapy for progression, it begs the question: why didn’t the other patients with progression stop first line treatment for that reason?

Answer: We agree that clarification is needed. These remaining patients actually progressed after TKI discontinuation. This was added in the revised version, Result section page 5.

Table 2 – combining “Adverse Events” with "patient request" makes conclusions difficult; please also clarify “scheduled treatment discontinuation or ongoing treatment” – as this would seem as though patients with ongoing treatment will be included in the reasons for discontinuation. The numbers don’t seem to reflect that, though. Please clarify.

Answer: We agree that these points require further clarification. Combining adverse events and patient request came from the fact that adverse events justifying the request of interruption by patients were qualified by the patients themselves as minimal. For clarification, we keep AE in the table and describe this in the discussion. Table 2 has been modified for clarification.

We also agree that « scheduled treatment discontinuation or ongoing treatment » is unclear. This was replaced by « patients in whom the treatment was not interrupted because of progression or AE, or patients in whom treatment is still ongoing at the time of the analysis » in the revised version in Table 2.

Line 183 – Please clarify how the two patients who started the same CSF1R inhibitor as second line therapy for either progression with symptoms or symptoms without progression would still count as second line therapy. At first glance, that would seem to be continuation on therapy if the antagonist is the same. Presumably, these are patients who completed treatment, demonstrated progression off treatment, and were re-started on therapy again. To include this scenario (relapse after completion of therapy) together with patients who progressed on therapy really confuses the overall analysis. This is a large limitation of the current study.

Answer: The reviewer is right: these are patients who completed treatment, demonstrated progression off treatment, and were re-started on therapy again. We respectfully would like to mention that this is generally reported as second line treatment in patients with advanced sarcoma receiving eg doxorubicin in first line interrupted after 6 courses. We applied the same logic here. Because few patients progressed during the treatment, this is actually the case for the majority of patients. We agree that this was not clear enough in the original version and have therefore amended in the Discussion section in this revised version. 

End of responses.

---

## [Decision Letter · Decision Letter 1]

5 Mar 2020

PONE-D-19-26498R1

Long term term follow-up of tyrosine kinase inhibitors treatments in inoperable or relapsing diffuse-type tenosynovial giant cell tumors (dTGCT).

PLOS ONE

Dear Dr. Blay,

Thank you for submitting your manuscript to PLOS ONE. After careful consideration, we feel that it has merit but does not fully meet PLOS ONE’s publication criteria as it currently stands. Therefore, we invite you to submit a revised version of the manuscript that addresses the points raised during the review process.

In response to the suggestions of the reviewers, please 1) rewrite to make it more obvious that this is a study of 39 patients who received either a TKI or an antibody targeting CSF-1R (this is not a study of all 117 patients referred to your center, nor of the 101 patients who had their diagnosis confirmed), and 2) revise the concluding paragraph to better clarify the "take home message" for clinicians.

We would appreciate receiving your revised manuscript by Apr 19 2020 11:59PM. To enhance the reproducibility of your results, we recommend that if applicable you deposit your laboratory protocols in protocols.io, where a protocol can be assigned its own identifier (DOI) such that it can be cited independently in the future. For instructions see: http://journals.plos.org/plosone/s/submission-guidelines#loc-laboratory-protocols

We look forward to receiving your revised manuscript.

Kind regards,

David M Loeb

Academic Editor

PLOS ONE

Reviewers' comments:

Reviewer's Responses to Questions

**Comments to the Author**

1. If the authors have adequately addressed your comments raised in a previous round of review and you feel that this manuscript is now acceptable for publication, you may indicate that here to bypass the “Comments to the Author” section, enter your conflict of interest statement in the “Confidential to Editor” section, and submit your "Accept" recommendation.

Reviewer #1: (No Response)

Reviewer #2: (No Response)

2. Is the manuscript technically sound, and do the data support the conclusions?

Reviewer #1: Yes

Reviewer #2: Partly

3. Has the statistical analysis been performed appropriately and rigorously? 

Reviewer #1: No

Reviewer #2: Yes

4. Have the authors made all data underlying the findings in their manuscript fully available?

Reviewer #1: Yes

Reviewer #2: Yes

5. Is the manuscript presented in an intelligible fashion and written in standard English?

Reviewer #1: Yes

Reviewer #2: No

6. Review Comments to the Author

Reviewer #1: Table 1 presents the clinical characteristics of these patients, at diagnosis. As of Jul 1st 2019, with a median follow-up since diagnosis of 69 months, none of these 101 patients have died and 1 only required an amputation. 62of these 101 patients did not receive TKI or Ab during the observation period. With a median follow-up of 16 months for this cohort, 4 documented progression and/or worsening symptoms were reported between 25 and 38months (not shown)Conversely, 39 (39%) have received so far first line CSF1R TKI as compassionate use (imatinib, nilotinib,...) other tyrosine kinase inhibitors or CSF1R Ab in early clinical trials (Table 1). The median duration of the first line treatment for these 39 patients was 7 months (range 1-30+). 35 of these 39 (89.7%) of patients stopped the treatment for another reason than volumetric progression. With a median follow-up of 30 months since TKI initiation, 15 (38%) presented a novel volumetric progression and/or worsening symptoms, 11 after treatment discontinuation. Tumor progression was reported in 13 of 15 (87%) and worsening symptoms only in n=2 (13%). Median time to progression (TTP) is not reached for these 39 patients: progression free rate was 56% at 30 months at the time of the analysis (Table 2 & Figure 1).

Is the manuscript technically sound, and do the data support the conclusions? (Answer options: Yes, No, Partly) Yes - particularly given that the paper is essentially a retrospective description of the responses to therapy. However, I feel the findings and the results are presented in a somewhat confusing manner. For example, it may be helpful to lay out n broad strokes how many patients ultimately responded to medical treatment overall, how many required 1st line therapy, 2 lines of therapy, etc. This would be a helpful take away message for clinicians and patients - which gets lost in the way the results are presented.

Has the statistical analysis been performed appropriately and rigorously? (Answer options: Yes, No, I don't know, N/A) Well - the statistics are really descriptive - although in methods they allude to comparison between subgroups and use of statistical software. Unless I am missing something, I think this is all descriptive and should be presented as such. There are no formal comparisons that I can see.

Reviewer #2: Some grammatical errors remain, which could be easily edited

Some additional edits may be necessary: "(imatinib, nilotinib,...)"

The manuscript is improved with the changes, though severe limitations remain. In the end, this is still a heterogeneous group of various therapies, in various locations, in a relatively small single center analysis.

The abstract and manuscript should clearly note that this is a study of 39 patients. Since the title suggests that this is an analysis of patients treated with tyrosine kinase inhibition, the abstract results section should lead with the 39 patients eligible, not the 101 screened. The first line of abstract results should read something like:

"Overall, 39 of 101 histologically confirmed dTGCT treated at our institution received at least one TKI." The small number of actually included patients for the primary analysis seems to be buried into the middle of paragraphs, rather than highlighted.

7. PLOS authors have the option to publish the peer review history of their article (what does this mean?). If published, this will include your full peer review and any attached files.

Reviewer #1: No

Reviewer #2: No

---

## [Author Response · Author response to Decision Letter 1]

26 Mar 2020

Dear Editor,

We would like to thank again the reviewers for their attention to this work and their helpful comments which helped us to improve the quality of our manuscript. 

Please find enclosed hereunder a revised version of the manuscript, making it explicit that this is a study on 39 patients with dTGCT. A revised version of the manuscript, tables and figures, with visible modifications and integrated modifications is downloaded on the PLoS One website. 

We of course remain at your disposition for any further questions you may have on this work. 

With kind regards,

Prof. JY Blay

---

## [Editor Report · Decision Letter 2]

28 Apr 2020

Long term term follow-up of tyrosine kinase inhibitors treatments in inoperable or relapsing diffuse-type tenosynovial giant cell tumors (dTGCT).

PONE-D-19-26498R2

Dear Dr. Blay,

We are pleased to inform you that your manuscript has been judged scientifically suitable for publication and will be formally accepted for publication once it complies with all outstanding technical requirements.

With kind regards,

David M Loeb

Academic Editor

PLOS ONE
---

## [Editor Report · Acceptance letter]

6 May 2020

PONE-D-19-26498R2 

Long term term follow-up of tyrosine kinase inhibitors treatments in inoperable or relapsing diffuse type tenosynovial giant cell tumors (dTGCT). 

Dear Dr. Blay:

I am pleased to inform you that your manuscript has been deemed suitable for publication in PLOS ONE. Congratulations! Your manuscript is now with our production department. 

With kind regards,

on behalf of

Dr. David M Loeb 

Academic Editor

PLOS ONE